# Identification of HLA-A*24:02-Restricted CTL Candidate Epitopes Derived from the Nonstructural Polyprotein 1a of SARS-CoV-2 and Analysis of Their Conservation Using the Mutation Database of SARS-CoV-2 Variants

Akira Takagi,[a]* Masanori Matsui[b]

[a]School of Medical Technology, Faculty of Health and Medical Care, Saitama Medical University, Yamane, Hidaka-city, Saitama, Japan
[b]Department of Microbiology, Faculty of Medicine, Saitama Medical University, Morohongo, Moroyama-cho, Iruma-gun, Saitama, Japan

**ABSTRACT** COVID-19 vaccines are currently being administered worldwide and playing a critical role in controlling the pandemic. They have been designed to elicit neutralizing antibodies against Spike protein of the original SARS-CoV-2, and hence they are less effective against SARS-CoV-2 variants with mutated Spike than the original virus. It is possible that novel variants with abilities of enhanced transmissibility and/or immunoevasion will appear in the near future and perfectly escape from vaccine-elicited immunity. Therefore, the current vaccines may need to be improved to compensate for the viral evolution. For this purpose, it may be beneficial to take advantage of CD8$^+$ cytotoxic T lymphocytes (CTLs). Several lines of evidence suggest the contribution of CTLs on the viral control in COVID-19, and CTLs target a wide range of proteins involving comparatively conserved nonstructural proteins. Here, we identified 22 HLA-A*24:02-restricted CTL candidate epitopes derived from the nonstructural polyprotein 1a (pp1a) of SARS-CoV-2 using computational algorithms, HLA-A*24:02 transgenic mice and the peptide-encapsulated liposomes. We focused on pp1a and HLA-A*24:02 because pp1a is relatively conserved and HLA-A*24:02 is predominant in East Asians such as Japanese. The conservation analysis revealed that the amino acid sequences of 7 out of the 22 epitopes were hardly affected by a number of mutations in the Sequence Read Archive database of SARS-CoV-2 variants. The information of such conserved epitopes might be useful for designing the next-generation COVID-19 vaccine that is universally effective against any SARS-CoV-2 variants by the induction of both anti-Spike neutralizing antibodies and CTLs specific for conserved epitopes.

**IMPORTANCE** COVID-19 vaccines have been designed to elicit neutralizing antibodies against the Spike protein of the original SARS-CoV-2, and hence they are less effective against variants. It is possible that novel variants will appear and escape from vaccine-elicited immunity. Therefore, the current vaccines may need to be improved to compensate for the viral evolution. For this purpose, it may be beneficial to take advantage of CD8$^+$ cytotoxic T lymphocytes (CTLs). Here, we identified 22 HLA-A*24:02-restricted CTL candidate epitopes derived from the nonstructural polyprotein 1a (pp1a) of SARS-CoV-2. We focused on pp1a and HLA-A*24:02 because pp1a is conserved and HLA-A*24:02 is predominant in East Asians. The conservation analysis revealed that the amino acid sequences of 7 out of the 22 epitopes were hardly affected by mutations in the database of SARS-CoV-2 variants. The information might be useful for designing the next-generation COVID-19 vaccine that is universally effective against any variants.

**KEYWORDS** SARS-CoV-2, COVID-19, CTL epitope, HLA-A*24:02, pp1a, vaccine, conserved epitope, variants

Address correspondence to Masanori Matsui, mmatsui@saitama-med.ac.jp.

*Present address: Akira Takagi, Biotechnology Research Division, Kohjin Bio Co., Ltd., 5-1-3 Chiyoda, Sakado city, Saitama, Japan.

The authors declare no conflict of interest.

Microbiology Spectrum

The severe acute respiratory syndrome coronavirus 2 (SARS-CoV-2) is the causative agent of the coronavirus disease 2019 (COVID-19) pandemic, which has resulted in more than 222 million infections and 4.6 million deaths around the world as of 10th September 2021. To bring the pandemic under control, a number of vaccine candidates are being developed at an unprecedented speed, and several of them are currently being administered all over the world. These include two mRNA vaccines of BNT162b2 (Pfizer/BioNTech) and mRNA-1273 (Moderna), and adenoviral-vectored vaccines such as ChAdOx1 nCoV-19 (AstraZeneca), Gam-COVID-Vac (Sputnik V, Gamaleya Research Institute), and Ad26.COV2.S (Janssen). Most of these vaccines have been designed to elicit neutralizing antibodies against the SARS-CoV-2 spike (S) protein that block the interaction between SARS-CoV-2 and the angiotensin-converting enzyme 2 on target cells, and thereby preventing SARS-CoV-2 infection. The Pfizer/BioNTech and Moderna mRNA vaccines showed 95% and 94.1% efficacy in preventing the onset of disease caused by SARS-CoV-2 (1, 2), respectively, whereas adenoviral-vectored vaccines demonstrated protection at a slightly lower but sufficient efficacy (3, 4). Therefore, it was initially speculated that these vaccines would put an end to the pandemic sooner or later. However, the recent emergence of various SARS-CoV-2 variants has made us realize that the initial idea was optimistic.

Although SARS-CoV-2 changes more slowly than most other RNA viruses because of a proofreading mechanism (5), its variants have continuously emerged in the circulating viruses. Since the fall of 2020, variant strains with enhanced transmissibility have been found in the United Kingdom (Alpha or B.1.1.7), South Africa (Beta or B.1.351) and Brazil (Gamma, B.1.1.28 or P.1). Because the vaccines were directed against the original SARS-CoV-2 virus that appeared in 2019, it was questioned whether the vaccines would quell SARS-CoV-2 variants. Although the impact of the first detected Alpha variant on anti-SARS-CoV-2 humoral immunity was found to be moderate (6–8), the Beta and Gamma variants significantly reduced susceptibility to neutralizing antibodies (8, 9). Particularly, the Beta was the most resistant to available monoclonal antibodies, convalescent and vaccinated sera (7, 8, 10–13). Fortunately, however, it was demonstrated that the BNT162b2 (Pfizer/BioNTech) and mRNA-1273 (Moderna) mRNA vaccines were highly effective against the Beta variant infection in Qatar, and people who had received two doses of this vaccine were almost completely protected from severe disease caused by the Beta variant (14, 15), suggesting that the current vaccines are still effective even against the Beta variant. In December 2020, another variant of concern, the Delta strain (B.1.617.2) has appeared in India. The Delta variant is more transmissible than the highly contagious Alpha variant (16), and this fastest strain has been expected to rapidly outcompete other variants and become the dominant lineage in many parts of the world (17). Furthermore, it was reported that viral loads in Delta infections were ~1,000 times higher than those in initial infections in early 2020 (18). It was also shown that unvaccinated individuals infected with the Delta were more likely to be hospitalized than unvaccinated people infected with the Alpha (19) although it is still unclear whether the Delta variant causes more severe illness than the previous strains. On the other hand, real-world data demonstrated that only modest differences in the vaccine efficiency were observed between the Delta and the Alpha after the receipt of two doses of vaccine (BNT162b2 or ChAdOx1 nCoV-19) (20), suggesting that two doses of the current vaccine may be effective against the Delta. However, the breakthrough infection with the Delta is often observed in fully vaccinated individuals. It has been proven that the vaccinated people with the Delta breakthrough infection may not develop severe disease but may have the potential to transmit SARS-CoV-2 to others as the same rates as those who are unvaccinated (21). Taken together, the current vaccines are likely to be still efficient for existing variants, including the Beta and Delta, but becoming less effective against the variants than the original virus. In addition to these four variants of concern, several variants of interest have emerged all over the world. Accordingly, it is possible that new variants with abilities of more enhanced transmissibility and/or immunoevasion will appear in the near future and perfectly escape from natural and vaccine-elicited immunity. Therefore, the current

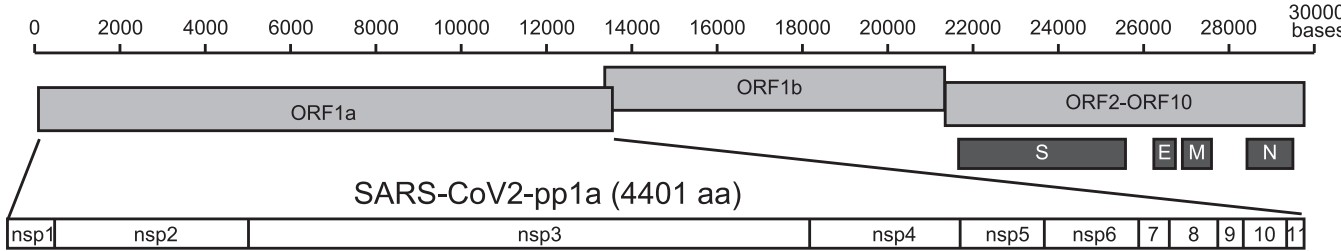

**FIG 1** The linear diagrams of the SARS-CoV-2 genome and the protein subunits of ORF1a. The SARS-CoV-2 genome consists of ORF1a, ORF1b, and ORF2-ORF10. S, E, M, and N represent spike, envelope, membrane, and nucleocapsid, respectively. The ORF1a polyprotein (pp1a) is composed of 11 nonstructural proteins, nsp1-nsp11.

vaccines may need to be improved to the next-generation vaccines in order to compensate for the viral evolution.

In general, CD8+ cytotoxic T lymphocytes (CTLs) play a crucial role for the clearance of virus as well as neutralizing antibodies in the viral infection. CTLs recognize virus-derived peptides in association with major histocompatibility complex class I (MHC-I) molecules on the surface of antigen-presenting cells and kill virus-infected target cells. In COVID-19, there was a greater proportion of SARS-CoV-2-specific CD8+ T cells in mild disease compared with severe case (22–25), suggesting a potential protective role of CD8+ T cell response. In fact, two persons with X-linked agammaglobulinemia recovered from pneumonia caused by the SARS-CoV-2 infection (26). In the virus-challenge experiment using rhesus macaques, depletion of CD8+ T cells in convalescent macaques that had been infected with SARS-CoV-2 partially abrogated the protective efficacy of natural immunity against rechallenge with SARS-CoV-2 (27), suggesting CD8+ T cells can contribute to virus control in COVID-19. The current mRNA vaccine and adenoviral-vectored vaccine elicit SARS-CoV-2 S protein-specific CD8+ CTLs as well as anti-S neutralizing antibodies (28), which might make these vaccines more efficient than inactivated and subunit vaccines. It is known that BNT162b2 mediates protection from severe disease as early as 10 days after prime vaccination, when neutralizing antibodies are hardly detectable. Since functional S-specific CD8+ T cells were shown to be already present at this early stage, CD8+ T cells were speculated to be the main mediators of the protection (29). Thus, several lines of evidence suggest the contribution of CTLs on the viral control in COVID-19, and therefore it may be beneficial to take advantage of CD8+ CTLs for the development of the next-generation vaccine. In addition, CTLs can target a wide range of proteins involving comparatively conserved nonstructural proteins. A novel vaccine with ability to elicit conserved epitope-specific CTLs may not be affected by mutations of various SARS-CoV-2 variants.

As shown in Fig. 1, the 5′-terminal two-thirds of the genome of SARS-CoV-2 are composed of the open reading frame 1a (ORF1a) and ORF1b. The ORF1a encodes the polyprotein 1a (pp1a) which is a largest protein composed of 11 nonstructural regulatory proteins (nsp1-11) in SARS-CoV-2. Due to its large size, it seems highly possible to find dominant epitopes in the pp1a. Saini et al. revealed that most of the immunodominant epitopes they identified belonged to the ORF1 region (30). In addition, it may be possible to identify conserved CTL epitopes in the pp1a because the ORF1 region is highly conserved within coronaviruses relative to structural proteins (31). From the above, we here attempted to identify conserved CTL epitopes derived from pp1a of SARS-CoV-2 using MHC-I transgenic mice. We focused on HLA-A*24:02-resctricted CTL epitopes because HLA-A*24:02 is relatively predominant in East Asians such as Japanese (32). This information might be useful for designing the next-generation COVID-19 vaccine that is universally effective against any SARS-CoV-2 variants by the induction of both anti-Spike neutralizing antibodies and CTLs specific for conserved epitopes.

## RESULTS

**Prediction of HLA-A*24:02-restricted CTL epitopes derived from SARS-CoV-2 pp1a.** To predict HLA-A*24:02-resctricted CTL epitopes derived from SARS-CoV-2 pp1a, we used a T-cell epitope database, SYFPEITHI (33). The top 80 epitopes in the database

**TABLE 1** HLA-A*24:02-restricted CTL candidate epitopes for the SARS-CoV-2 pp1a

| Name[a] | Sequence | Rank by algorithm[b] | | | | Name | Sequence | Rank by algorithm | | | |
|---|---|---|---|---|---|---|---|---|---|---|---|
| | | Number 1 | Number 2 | Number 3 | Number 4 | | | Number 1 | Number 2 | Number 3 | Number 4 |
| pp1a-620 | VYEKLKPVL | A | B | A | C | pp1a-3606 | FYENAFLPF | B | B | A | A |
| pp1a-954 | DYQGKPLEF | A | A | A | A | pp1a-3907 | AFEKMVSLL | B | C | C | D |
| pp1a-1089 | DYIATNGPL | A | D | A | C | pp1a-4090 | TYASALWEI | B | A | B | A |
| pp1a-1182 | LYDKLVSSF | A | A | B | A | pp1a-4378 | GYGCSCDQL | B | D | A | D |
| pp1a-1255 | LYIDINGNL | A | B | A | B | pp1a-486 | AFVETVKGL | C | D | C | D |
| pp1a-1733 | SYLFQHANL | A | A | A | B | pp1a-640 | EFLRDGWEI | C | C | D | C |
| pp1a-1813 | QYELKHGTF | A | B | B | B | pp1a-708 | TFVTHSKGL | C | D | C | D |
| pp1a-1845 | LYCIDGALL | A | B | A | C | pp1a-1137 | NFNQHEVLL | C | C | C | D |
| pp1a-2167 | NYMPYFFTL | A | A | A | A | pp1a-1552 | TFDNLKTLL | C | C | C | D |
| pp1a-2330 | AYILFTRFF | A | A | B | A | pp1a-1634 | YYHTTDPSF | C | A | B | A |
| pp1a-2436 | VYANGGKGF | A | A | B | C | pp1a-1906 | YFTEQPIDL | C | C | C | D |
| pp1a-3104 | VYSVIYLYL | A | A | A | A | pp1a-1929 | KFVCDNIKF | C | B | C | C |
| pp1a-3108 | IYLYLTFYL | A | B | A | A | pp1a-1936 | KFADDLNQL | C | B | B | D |
| pp1a-3114 | FYLTNDVSF | A | A | B | A | pp1a-1971 | DYKHYTPSF | C | A | B | B |
| pp1a-3159 | NYLKRRVVF | A | A | B | A | pp1a-1978 | SFKKGAKLL | C | C | C | D |
| pp1a-3684 | MYASAVVLL | A | A | A | A | pp1a-2222 | NFSKLINII | C | C | D | D |
| pp1a-3792 | CYFGLFCLL | A | C | A | B | pp1a-2232 | WFLLLSVCL | C | D | C | C |
| pp1a-3812 | DYLVSTQEF | A | A | A | A | pp1a-2320 | AFGLVAEWF | C | B | D | C |
| pp1a-3821 | RYMNSQGLL | A | B | A | B | pp1a-2781 | LFVAAIFYL | C | D | C | C |
| pp1a-4226 | KYLYFIKGL | A | B | A | B | pp1a-2826 | CFANKHADF | C | C | D | C |
| pp1a-96 | QYGRSGETL | B | B | A | C | pp1a-3030 | MFTPLIQPI | C | C | D | D |
| pp1a-135 | SYGADLKSF | B | A | B | B | pp1a-3084 | LFLMSFTVL | C | D | C | C |
| pp1a-616 | IFGTVYEKL | B | B | C | B | pp1a-3610 | AFLPFAMGI | C | C | D | D |
| pp1a-634 | KFKEGVEFL | B | B | C | D | pp1a-3627 | MFVKHKHAF | C | B | D | C |
| pp1a-677 | TFFKLVNKF | B | A | D | B | pp1a-3752 | MFLARGIVF | C | B | D | B |
| pp1a-835 | GYKSVNITF | B | A | B | A | pp1a-3837 | AFKLNIKLL | C | C | C | D |
| pp1a-1247 | KFLTENLLL | B | B | B | B | pp1a-4229 | YFIKGLNNL | C | B | C | D |
| pp1a-1451 | GYVTHGLNL | B | B | A | C | pp1a-265 | TFNGECPNF | D | B | D | D |
| pp1a-1515 | SYSGQSTQL | B | B | A | B | pp1a-335 | DFVKATCEF | D | B | D | C |
| pp1a-1536 | YYTSNPTTF | B | A | B | A | pp1a-1352 | AFYILPSII | D | C | D | C |
| pp1a-1899 | YYKKDNSYF | B | A | B | B | pp1a-1417 | DYGARFYFY | D | C | D | B |
| pp1a-2002 | TYKPNTWCI | B | A | B | A | pp1a-1543 | TFHLDGEVI | D | D | D | D |
| pp1a-2338 | FYVLGLAAI | B | B | B | C | pp1a-2333 | LFTRFFYVL | D | C | C | C |
| pp1a-2601 | TFNVPMEKL | B | B | C | D | pp1a-2350 | FFSYFAVHF | D | C | D | B |
| pp1a-2779 | VFLFVAAIF | B | B | C | B | pp1a-2457 | TFCAGSTFI | D | D | D | D |
| pp1a-2931 | PYCYDTNVL | B | C | C | B | pp1a-2590 | MFDAYVNTF | D | A | D | B |
| pp1a-2953 | RYVLMDGSI | B | C | B | B | pp1a-2717 | DFMSLSEQL | D | C | C | C |
| pp1a-3010 | YYRSLPGVF | B | B | B | A | pp1a-3137 | PFWITIAYI | D | D | D | D |
| pp1a-3153 | FYWFFSNYL | B | B | A | B | pp1a-3396 | NFTIKGSFL | D | D | C | D |
| pp1a-3249 | LYQPPQTSI | B | A | B | C | pp1a-3788 | YFCTCYFGL | D | D | C | B |

[a]Number in the peptide name shows the 1st amino acid position of each peptide in the SARS-CoV-2 pp1a.
[b]Algorithm number 1, SYFPEITHI; number 2, IEDB; number 3, ProPred-I; number 4, NetCTL.
Scores of predicted peptides were assessed by classifying into four ranks (A, Excellent; B, Very good; C, Good; D, Poor) (SYFPEITHI, A ≥ 22, 20 ≤ B ≤ 21, 18 ≤ C ≤ 19, D =17; IEDB, A < 0.1, 0.1 ≤ B < 0.5, 0.5 ≤ C < 1, D ≥ 1; ProPred-I, A ≥ 160, 50 ≤ B < 160, 20 ≤ C < 50, D < 20; NetCTL, A ≥ 1.70, 1.19 ≤ B < 1.70, 0.90 ≤ C < 1.19, D < 0.90).

were selected and were synthesized into 9-mer peptides (Table 1). These epitopes were also evaluated by other three programs, IEDB (34), ProPred-1 (35), and NetCTL (36) (Table 1). Scores of the 80 epitopes in the four programs were assessed by classifying into four ranks (A: Excellent; B: Very good; C: Good; D: Poor) (SYFPEITHI: A ≥ 22, 20 ≤ B ≤ 21, 18 ≤ C ≤ 19, D = 17; IEDB: A < 0.1, 0.1 ≤ B < 0.5, 0.5 ≤ C < 1, D ≥ 1; ProPred-I: A ≥ 160, 50 ≤ B < 160, 20 ≤ C < 50, C < 20; NetCTL: A ≥ 1.70, 1.19 ≤ B < 1.70, 0.90 ≤ C < 1.19, C < 0.90) (Table 1). As shown in Table 1, the rank of each epitope was not always the same in the four programs, suggesting that multiple programs are needed to successfully predict CTL epitopes.

Eighty peptides were investigated for their binding affinities to HLA-A*24:02 molecules using TAP2-deficient RMA-S-HHD-A24 cells. Since the half-maximal binding level (BL$_{50}$) value of a positive-control peptide, Influenza PA$_{130-138}$ (37) was 2.4 $\mu$M, we defined an extremely high binder with a BL$_{50}$ value below 1.0 $\mu$M, a high binder with a BL$_{50}$ value ranging from 1 to 10 $\mu$M, a medium binder with a BL$_{50}$ value ranging from

**TABLE 2** Binding affinities of predicted SARS-CoV-2 pp1a peptides to HLA-A*24:02[a]

| Extremely high binders | | | | | |
| --- | --- | --- | --- | --- | --- |
| Name | Sequence | $BL_{50}$ | Name | Sequence | $BL_{50}$ |
| pp1a-835 | GYKSVNITF | 0.1 ± 0.0 | pp1a-1536 | YYTSNPTTF | 0.05 ± 0.00 |
| pp1a-1634 | YYHTTDPSF | 0.04 ± 0.00 | pp1a-1899 | YYKKDNSYF | 0.3 ± 0.0 |
| pp1a-2330 | AYILFTRFF | 0.8 ± 0.1 | pp1a-2338 | FYVLGLAAI | 0.7 ± 0.1 |
| pp1a-3104 | VYSVIYLYL | 0.5 ± 0.0 | pp1a-3114 | FYLTNDVSF | 0.01 ± 0.00 |
| pp1a-3606 | FYENAFLPF | 0.02 ± 0.01 | pp1a-3684 | MYASAVVLL | 0.4 ± 0.1 |
| pp1a-3812 | DYLVSTQEF | 0.9 ± 0.1 | | | |
| **High binders** | | | | | |
| Name | Sequence | $BL_{50}$ | Name | Sequence | $BL_{50}$ |
| pp1a-265 | TFNGECPNF | 1.7 ± 0.4 | pp1a-1182 | LYDKLVSSF | 1.0 ± 0.3 |
| pp1a-1733 | SYLFQHANL | 3.0 ± 0.4 | pp1a-2350 | FFSYFAVHF | 1.4 ± 0.1 |
| pp1a-2590 | MFDAYVNTF | 7.7 ± 0.6 | pp1a-2779 | VFLFVAAIF | 8.6 ± 0.4 |
| pp1a-2931 | PYCYDTNVL | 6.8 ± 1.0 | pp1a-3153 | FYWFFSNYL | 3.9 ± 1.3 |
| pp1a-3249 | LYQPPQTSI | 2.1 ± 1.2 | pp1a-3821 | RYMNSQGLL | 7.1 ± 5.0 |
| **Medium binders** | | | | | |
| Name | Sequence | $BL_{50}$ | Name | Sequence | $BL_{50}$ |
| pp1a-634 | KFKEGVEFL | 79.6 ± 6.6 | pp1a-677 | TFFKLVNKF | 55.9 ± 3.3 |
| pp1a-1137 | NFNQHEVLL | 55.3 ± 2.0 | pp1a-1247 | KFLTENLLL | 33.4 ± 1.9 |
| pp1a-1255 | LYIDINGNL | 40.4 ± 8.6 | pp1a-1352 | AFYILPSII | 64.4 ± 9.2 |
| pp1a-1417 | DYGARFYFY | 16.1 ± 0.8 | pp1a-1515 | SYSGQSTQL | 45.8 ± 10.5 |
| pp1a-1845 | LYCIDGALL | 38.5 ± 8.2 | pp1a-1929 | KFVCDNIKF | 61.7 ± 1.6 |
| pp1a-1971 | DYKHYTPSF | 58.4 ± 11.7 | pp1a-2953 | RYVLMDGSI | 28.1 ± 1.5 |
| pp1a-3752 | MFLARGIVF | 24.2 ± 1.5 | pp1a-3792 | CYFGLFCLL | 16.0 ± 3.8 |
| pp1a-4229 | YFIKGLNNL | 32.1 ± 1.9 | | | |
| **Low binders** | | | | | |
| Name | Sequence | $BL_{50}$ | Name | Sequence | $BL_{50}$ |
| pp1a-96 | QYGRSGETL | 124.5 ± 10.9 | pp1a-135 | SYGADLKSF | 103.8 ± 17.7 |
| pp1a-335 | DFVKATCEF | 127.2 ± 3.8 | pp1a-486 | AFVETVKGL | 277.9 ± 49.8 |
| pp1a-616 | IFGTVYEKL | 122.2 ± 18.0 | pp1a-620 | VYEKLKPVL | 263.6 ± 21.4 |
| pp1a-640 | EFLRDGWEI | ND | pp1a-708 | EFLRDGWEI | ND |
| pp1a-954 | DYQGKPLEF | 126.2 ± 23.9 | pp1a-1089 | DYIATNGPL | ND |
| pp1a-1451 | GYVTHGLNL | 109.2 ± 8.9 | pp1a-1543 | TFHLDGEVI | 100.2 ± 6.8 |
| pp1a-1552 | TFDNLKTLL | 96.1 ± 1.6 | pp1a-1813 | QYELKHGTF | 140.7 ± 8.0 |
| pp1a-1906 | YFTEQPIDL | 113.1 ± 9.4 | pp1a-1936 | KFADDLNQL | 42.4 ± 6.1 |
| pp1a-1978 | SFKKGAKLL | ND | pp1a-2002 | TYKPNTWCI | ND |
| pp1a-2167 | NYMPYFFTL | ND | pp1a-2222 | NFSKLINII | 181.5 ± 32.5 |
| pp1a-2232 | WFLLLSVCL | 279.4 ± 12.6 | pp1a-2320 | AFGLVAEWF | 135.5 ± 39.1 |
| pp1a-2333 | LFTRFFYVL | ND | pp1a-2436 | VYANGGKGF | ND |
| pp1a-2457 | TFCAGSTFI | 127.1 ± 20.2 | pp1a-2601 | TFNVPMEKL | ND |
| pp1a-2717 | DFMSLSEQL | ND | pp1a-2781 | LFVAAIFYL | 182.6 ± 8.2 |
| pp1a-2826 | CFANKHADF | ND | pp1a-3010 | YYRSLPGVF | 122.1 ± 18.8 |
| pp1a-3030 | MFTPLIQPI | 251.1 ± 88.7 | pp1a-3084 | LFLMSFTVL | 100.4 ± 7.6 |
| pp1a-3108 | IYLYLTFYL | 100.5 ± 13.8 | pp1a-3137 | PFWITIAYI | 103.3 ± 8.4 |
| pp1a-3159 | NYLKRRVVF | 114.3 ± 5.3 | pp1a-3396 | NFTIKGSFL | 134.0 ± 7.6 |
| pp1a-3610 | AFLPFAMGI | 182.6 ± 8.2 | pp1a-3627 | MFVKHKHAF | 240.9 ± 33.6 |
| pp1a-3788 | YFCTCYFGL | ND | pp1a-3837 | AFKLNIKLL | ND |
| pp1a-3907 | AFEKMVSLL | 113.9 ± 3.1 | pp1a-4090 | TYASALWEI | ND |
| pp1a-4226 | KYLYFIKGL | 109.3 ± 1.3 | pp1a-4378 | GYGCSCDQL | 137.9 ± 9.9 |

[a]Data of peptide binding assays are shown as $BL_{50}$, indicating a concentration ($\mu$M) of each peptide that yields the 50% relative binding as shown in the materials and methods. Experiments were performed in triplicate and repeated twice with similar results. Data are given as mean values ± SD. Extremely high binders, $BL_{50} < 1.0\,\mu$M; High binders, $1\,\mu$M $\leq BL_{50} < 10\,\mu$M; Medium binders, $10\,\mu$M $\leq BL_{50} < 80\,\mu$M; Low binders, $BL_{50} \geq 80\,\mu$M or ND (not detected).

10 to 80 $\mu$M, and a low binder with a $BL_{50}$ value above 80 $\mu$M. Among 80 peptides, 11peptides and 10 peptides were extremely high binders and high binders, respectively, while 15 peptides were medium binders (Table 2). The remaining 44 peptides demonstrated low binding affinities or no binding to HLA-A*24:02 (Table 2). Comparison of the peptide binding affinity and the peptide rank in the 4 algorithms (Table 3) revealed that A-ranked peptides did not always show the high level of the

**TABLE 3** Comparison between the peptide binding affinity and the rank[a] of peptides in the 4 algorithms

| Algorithm | Rank | Extremely high BL$_{50}$ < 1.0 ($\mu$M) | High binder 1 ≤ BL$_{50}$ < 10 | Medium binder 10 ≤ BL$_{50}$ < 80 | Low binder BL$_{50}$ ≥ 80 |
|---|---|---|---|---|---|
| SYFPEITHI | A | 5/80 (6.3%) | 3/80 (3.8%) | 3/80 (3.8%) | 9/80 (11.3%) |
| | B | 5/80 (6.3%) | 4/80 (5.0%) | 5/80 (6.3%) | 10/80 (10.0%) |
| | C | 1/80 (1.3%) | 0/80 (0%) | 5/80 (6.3%) | 17/80 (21.3%) |
| | D | 0/80 (0%) | 3/80 (3.8%) | 2/80 (2.5%) | 8/80 (10.0%) |
| IEDB | A | 9/80 (11.3%) | 4/80 (5.0%) | 2/80 (2.5%) | 7/80 (8.8%) |
| | B | 2/80 (2.5%) | 4/80 (5.0%) | 8/80 (10.0%) | 13/80 (16.3%) |
| | C | 0/80 (0%) | 2/80 (2.5%) | 5/80 (6.3%) | 12/80 (15.0%) |
| | D | 0/80 (0%) | 0/80 (0%) | 0/80 (0%) | 12/80 (15.0%) |
| ProPred-I | A | 4/80 (5.0%) | 3/80 (3.8%) | 4/80 (5.0%) | 9/80 (11.3%) |
| | B | 7/80 (8.8%) | 2/80 (2.5%) | 3/80 (3.8%) | 8/80 (10.0%) |
| | C | 0/80 (0%) | 2/80 (2.5%) | 4/80 (5.0%) | 16/80 (20.0%) |
| | D | 0/80 (0%) | 3/80 (3.8%) | 4/80 (5.0%) | 11/80 (13.8%) |
| NetCTL | A | 9/80 (11.3%) | 1/80 (1.3%) | 0/80 (0%) | 7/80 (8.8%) |
| | B | 1/80 (1.3%) | 7/80 (8.8%) | 9/80 (11.3%) | 5/80 (6.3%) |
| | C | 1/80 (1.3%) | 1/80 (1.3%) | 3/80 (3.8%) | 15/80 (18.8%) |
| | D | 0/80 (0%) | 1/80 (1.3%) | 3/80 (3.8%) | 17/80 (21.3%) |

[a]Rank: peptides were classified into four ranks (A, Excellent; B, Very good; C, Good; D, Poor) in each of the four algorithms (SYFPEITHI, IEDB, ProPred-I, NetCTL).

peptide binding affinity to HLA-A*24:02. On the other hand, none of d-ranked peptides were classified into the extremely high group. When comparing the four programs in the prediction of extremely high binders and high binders, the IEDB program was likely to estimate them most accurately (Fig. 2).

In the following experiments, 36 peptides involving extremely high, high, and medium binders were chosen to investigate their abilities of peptide-specific CTL induction.

**Induction of SARS-CoV-2 pp1a-specific CD8+ T cell responses in HLA-A*24:02 transgenic mice immunized with liposomal peptides.** The 36 peptides were randomly divided into 6 groups. Six peptides in each group were mixed and encapsulated into liposomes as described in the Materials and Methods. HLA-A*24:02 transgenic mice were then subcutaneously (s.c.) immunized four times at a 1-week interval with peptide-encapsulated liposomes together with CpG adjuvant. 1 week later, spleen cells of immunized mice were prepared, stimulated in vitro with a relevant peptide for 5 h, and stained for their expression of cell-surface CD8 and intracellular interferon gamma (IFN-$\gamma$). As shown in Fig. 3A, it was demonstrated that significant numbers of IFN-$\gamma$-producing CD8+ T cells were detected in mice immunized with 22 liposomal peptides, including pp1a-265, -634, -835, -1182, -1255, -1417, -1845, -1899, -2330, -2338, -2350, -2590, -2779, -3104,

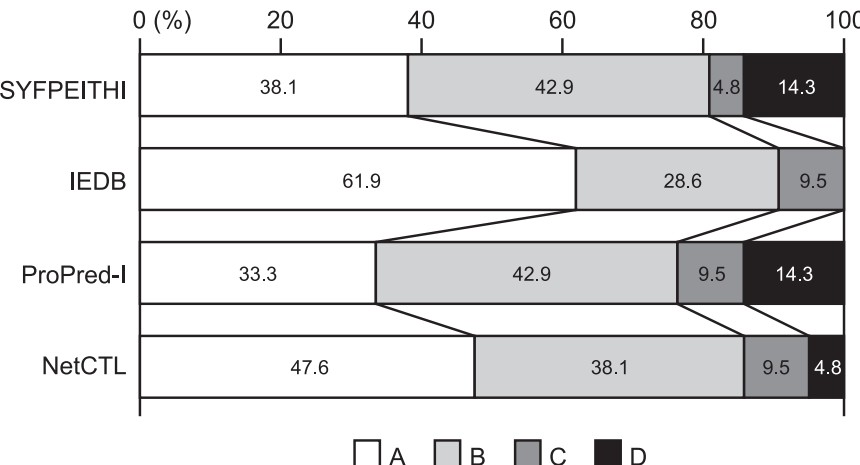

**FIG 2** Percentages of peptides ranked from A to D (A, Excellent; B, Very good; C, Good; D, Poor) by each algorithm in the sum of extremely high peptides and high binder peptides.

## A

| ICS | Count | pp1a- |
|-----|-------|-------|
| 1% ≤ | 5 (13.9%) | 265, 1255, 2330, 3104, 3792 |
| 0.5-1.0% | 7 (19.4%) | 634, 835, 1845, 2590, 3153 |
| | | 3249, 3606 |
| 0.1-0.5% | 10 (27.8%) | 1182, 1417, 1899, 2338, 2350 |
| | | 2779, 3114, 3684, 3752, 4229 |
| ND | 14 (38.9%) | 677, 1137, 1247, 1352, 1515 |
| | | 1536, 1634, 1733, 1929, 1971 |
| | | 2931, 2953, 3812, 3821 |

## B

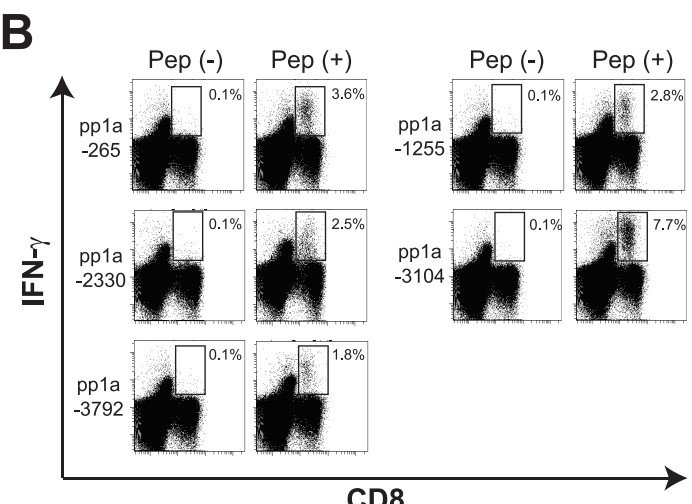

**FIG 3** Intracellular IFN-$\gamma$ staining (ICS) of CD8$^+$ T cells stimulated with peptides derived from SARS-CoV-2 pp1a. After HLA-A*24:02 transgenic mice were immunized with liposomal peptides derived from SARS-CoV-2 pp1a, spleen cells were stimulated with or without a relevant peptide for 5 h. Cells were stained for their surface expression of CD8 and their intracellular expression of IFN-$\gamma$. (A) Values of ICS show the relative percentages of IFN-$\gamma^+$ cells in CD8$^+$ T cells which were calculated by subtracting the % of IFN-$\gamma^+$ cells in CD8$^+$ T cells without a peptide from the % of IFN-$\gamma^+$ cells in CD8$^+$ T cells with a relevant peptide. Thirty-six peptides tested were divided into 4 groups with ICS values of 1% or higher, 0.5–1%, 0.1–0.5%, and ND (not detected). (B) Representative flow cytometry histograms are shown. Numbers shown indicate the percentages of intracellular IFN-$\gamma^+$ cells within CD8$^+$ T cells stimulated with (+) or without (-) a relevant peptide. The data shown are representative of three independent experiments. Three to five mice per group were used in each experiment, and spleen cells of mice per group were pooled.

-3114, -3153, -3249, -3606, -3684, -3752, -3792, and -4229. These data indicated that the 22 peptides were HLA-A*24:02-restricted CTL candidate epitopes derived from SARS-CoV-2 pp1a. However, the induction level of IFN-$\gamma$-producing CD8$^+$ T cells varied among the 22 peptides. Five peptides, including pp1a-265, -1255, -2330, -3104, and -3792 elicited high percentages of intracellular IFN-$\gamma^+$ cells in CD8$^+$ T cells, ranging from 1.8% to 7.7% (Fig. 3A and B), whereas the other 17 peptides induced medium (0.5–1%) or low percentages (0.1–0.5%) of IFN-$\gamma^+$ CD8$^+$ T cells (Fig. 3A). When comparing between the data of ICS and the peptide binding affinity (Table 4), it was shown that all of extremely high binders did not elicit IFN-$\gamma$ producing CD8$^+$ T cells and two medium binder peptides activated high percentages of intracellular IFN-$\gamma^+$ cells in CD8$^+$ T cells. However, the

**TABLE 4** Correlation between the peptide binding affinity and the peptide immunogenicity

| ICS[a] | Ext. high binder $BL_{50} < 1.0\ (\mu M)$ | High binder $1.0 \le BL_{50} < 10$ | Medium binder $10 \le BL_{50} < 80$ |
|---|---|---|---|
| 1% ≤ | 3 (27.3%) | 0 (0%) | 2 (13.3%) |
| 0.5–1% | 2 (18.2%) | 3 (30.0%) | 2 (13.3%) |
| 0.1 to 0.5% | 4 (36.4%) | 3 (30.0%) | 3 (20.0%) |
| ND | 2 (18.2%) | 4 (40.0%) | 8 (53.3%) |
| | | | |
| Total No. | 11 | 10 | 15 |

[a]Intracellular cytokine staining (ICS): The relative percentages of IFN-$\gamma^+$ cells in CD8$^+$ T cells which were calculated by subtracting the % of IFN-$\gamma^+$ cells in CD8$^+$ T cells without a peptide from the % of IFN-$\gamma^+$ cells in CD8$^+$ T cells with a relevant peptide. ND, not detected.

proportion of extremely high binder peptides that induced IFN-$\gamma$ producing CD8$^+$ T cells was higher than that of medium binder peptides (Table 4), confirming that the peptide binding affinity to HLA class I molecules is closely associated with the induction of peptide-specific CTLs.

**Conservation analysis of CTL epitopes in the database of SARS-CoV-2 variants.** We next investigated whether the 22 candidate epitopes were mutated in various SARS-CoV-2 variants. To do this, we utilized the National Center for Biotechnology Information (NCBI) Virus database (https://www.ncbi.nlm.nih.gov) (38), in which they provide us data-sets of mutations in the Sequence Read Archive (SRA) records of SARS-CoV-2 variants. In the database, the nucleotide and amino acid sequences of variants in SRA records were aligned for comparison with those of the original strain, Wuhan-Hu-1 (NCBI Reference Sequence: NC_045512.2). In the SRA mutation data, the most frequent, nonsynonymous amino acid change was the mutation from D to G at position 614 (D614G) in the S protein, and the total count of D614G across the database was 615,601 in 924,785 SRA runs (Frequency per run: 66.6%) available as of 23rd August 2021. To investigate the conservation of the 22 epitopes, we counted the total number of nonsynonymous amino acid substitutions present in the 9-mer amino acid sequence of each epitope that were found in a number of SRA sequencing data of SARS-CoV-2 variants in 924,785 SRA runs. It was discovered that all of those epitopes had more or less amino acid substitutions in their amino acid sequences (Table 5 & Fig. 4), indicating none of them were fully conserved throughout all of the available SRA data. However, there were seven epitopes with low counts of total mutations present in their 9-mer amino acid sequences, indicating that the amino acid sequences of the seven epitopes were hardly affected by a number of mutations in the SRA database (Table 5). The 7 epitopes were pp1a-835 (559 count in 924,785 SRA runs; Frequency per run: 0.06%), -1417 (245; Frequency: 0.03%), -1899 (531; Frequency: 0.06%), -2590 (611; Frequency: 0.07%), -3104 (142; Frequency: 0.02%), -3792 (336; Frequency: 0.04%) and -4229 (83; Frequency: 0.01%) (Table 5). The number of mutations at each amino acid position in the 9-mer amino acid sequence of an epitope was shown in Fig. 4. In contrast, numbers of amino acid changes in the epitope sequences were very high in some other epitopes, including pp1a-265 (53,049; Frequency per run: 5.74%), -2779 (18,819; Frequency: 2.03%), -3249 (126,956; Frequency: 13.73%), and -3606 (19,733; Frequency: 2.13%) (Table 5).

It was then determined which of the relatively conserved top 4 epitopes, namely, pp1a-1417, -3104, -3792, and -4229, was most dominant in the induction of pp1a-specific CTLs. The same amounts of the 4 peptide solutions at an equal concentration were mixed together and encapsulated into liposomes. Eight mice were immunized with the liposomes containing the peptide mixture. 1 week later, spleen cells were incubated with each of the 4 peptides for 5 h, and the ICS assay was performed. It was found that pp1a-3104 was far superior to all other peptides in the induction of peptide-specific IFN-$\gamma^+$ CD8$^+$ T cells (Fig. 5A). We also examined the peptide-specific induction of CD8$^+$ T cells expressing a degranulation marker, CD107a. As shown in Fig. 5B, pp1a-3104 was statistically predominant over pp1a-1417 and -4229 for the CD107a

**TABLE 5** Count of total nonsynonymous amino acid changes in each of the 22 HLA-A*24:02-restricted, pp1a-specific CTL candidate epitopes

| Name | Sequence | Protein | Position[a] | Count[b] |
|---|---|---|---|---|
| pp1a-265 | TFNGECPNF | nsp2 | 85-93 | 53049 (5.74%) |
| pp1a-634 | KFKEGVEFL | nsp2 | 454-462 | 1227 (0.13%) |
| pp1a-835 | GYKSVNITF | nsp3 | 17-25 | 559 (0.06%) |
| pp1a-1182 | LYDKLVSSF | nsp3 | 364-372 | 4016 (0.43%) |
| pp1a-1255 | LYIDINGNL | nsp3 | 437-445 | 1089 (0.12%) |
| pp1a-1417 | DYGARFYFY | nsp3 | 599-607 | 245 (0.03%) |
| pp1a-1845 | LYCIDGALL | nsp3 | 1027-1035 | 1394 (0.15%) |
| pp1a-1899 | YYKKDNSYF | nsp3 | 1081-1089 | 531 (0.06%) |
| pp1a-2330 | AYILFTRFF | nsp3 | 1512-1520 | 1066 (0.12%) |
| pp1a-2338 | FYVLGLAAI | nsp3 | 1520-1528 | 2614 (0.28%) |
| pp1a-2350 | FFSYFAVHF | nsp3 | 1532-1540 | 1041 (0.11%) |
| pp1a-2590 | MFDAYVNTF | nsp3 | 1772-1780 | 611 (0.07%) |
| pp1a-2779 | VFLFVAAIF | nsp4 | 16-24 | 18819 (2.03%) |
| pp1a-3104 | VYSVIYLYL | nsp4 | 341-349 | 142 (0.02%) |
| pp1a-3114 | FYLTNDVSF | nsp4 | 351-359 | 1425 (0.15%) |
| pp1a-3153 | FYWFFSNYL | nsp4 | 390-398 | 3252 (0.35%) |
| pp1a-3249 | LYQPPQTSI | nsp4 | 486-494 | 126956 (13.73%) |
| pp1a-3606 | FYENAFLPF | nsp6 | 37-45 | 19733 (2.13%) |
| pp1a-3684 | MYASAVVLL | nsp6 | 115-123 | 943 (0.10%) |
| pp1a-3752 | MFLARGIVF | nsp6 | 183-191 | 4559 (0.49%) |
| pp1a-3792 | CYFGLFCLL | nsp6 | 223-231 | 336 (0.04%) |
| pp1a-4229 | YFIKGLNNL | nsp9 | 89-97 | 83 (0.01%) |

[a]Amino acid position in each nonstructural protein.
[b]Count of the total amino acid substitutions present in the 9-mer amino acid sequence of each HLA-A*24:02-restricted, pp1a-specific CTL candidate epitope that were found in the SRA database of SARS-CoV-2 variants. Percentage in parenthesis indicates the mutation frequency per SRA run.

induction of CD8+ T cells. Thus, it was found that pp1a-3104 was the most prominent HLA-A*24:02-restricted CTL epitope among the conserved top 4 epitopes.

## DISCUSSION

All of the current available COVID-19 vaccines have been directed against the S protein of the original SARS-CoV-2, and therefore they are less effective against some variants with mutated S such as the Beta and Delta strains than the original virus. Our concern is that SARS-CoV-2 is currently under evolution and various variants are appearing one after another. One day soon, new mutant strains that perfectly evade the immunity generated by the vaccines may emerge. To develop a next-generation vaccine to compensate for the viral evolution, it may be beneficial to take advantage of CTLs because they can target a wide range of SARS-CoV-2-derived proteins, involving comparatively conserved nonstructural proteins.

Here, we have identified 22 HLA-A*24:02-restricted CTL candidate epitopes derived from SARS-CoV-2 pp1a using HLA-A*24:02 transgenic mice. The pp1a is a large polyprotein consisting of 4,401 amino acids that may be relatively conserved compared to structural proteins such as the S protein (31). Furthermore, Tarke et al. demonstrated that most of T cell epitopes they identified were conserved across the Alpha, Beta, Gamma, and Epsilon (CAL.20C) variants, and the impact of the four variants on the total CD8+ T cell reactivity in vaccinated individuals was negligible (39). Hence, we first thought it might be possible to find pp1a-derived epitopes that were fully conserved across a number of the existing SARS-CoV-2 variants. Unfortunately, none of the 22 epitopes we identified were found to be completely conserved throughout vast amounts of the SRA data in the NCBI Virus database. This is understandable because most (73.3%) of the 4,401 amino acids that make up the pp1a have nonsynonymous amino acid substitutions found in the SARS-CoV-2 SRA data. As shown in Table 5, however, seven epitopes, including pp1a-835, -1417, -1899, -2590, -3104, -3792, and -4299 were relatively conserved due to low counts of total mutations and minimum mutation frequencies of less than 0.1% in their amino acid sequences. Of note, pp1a-3104 was

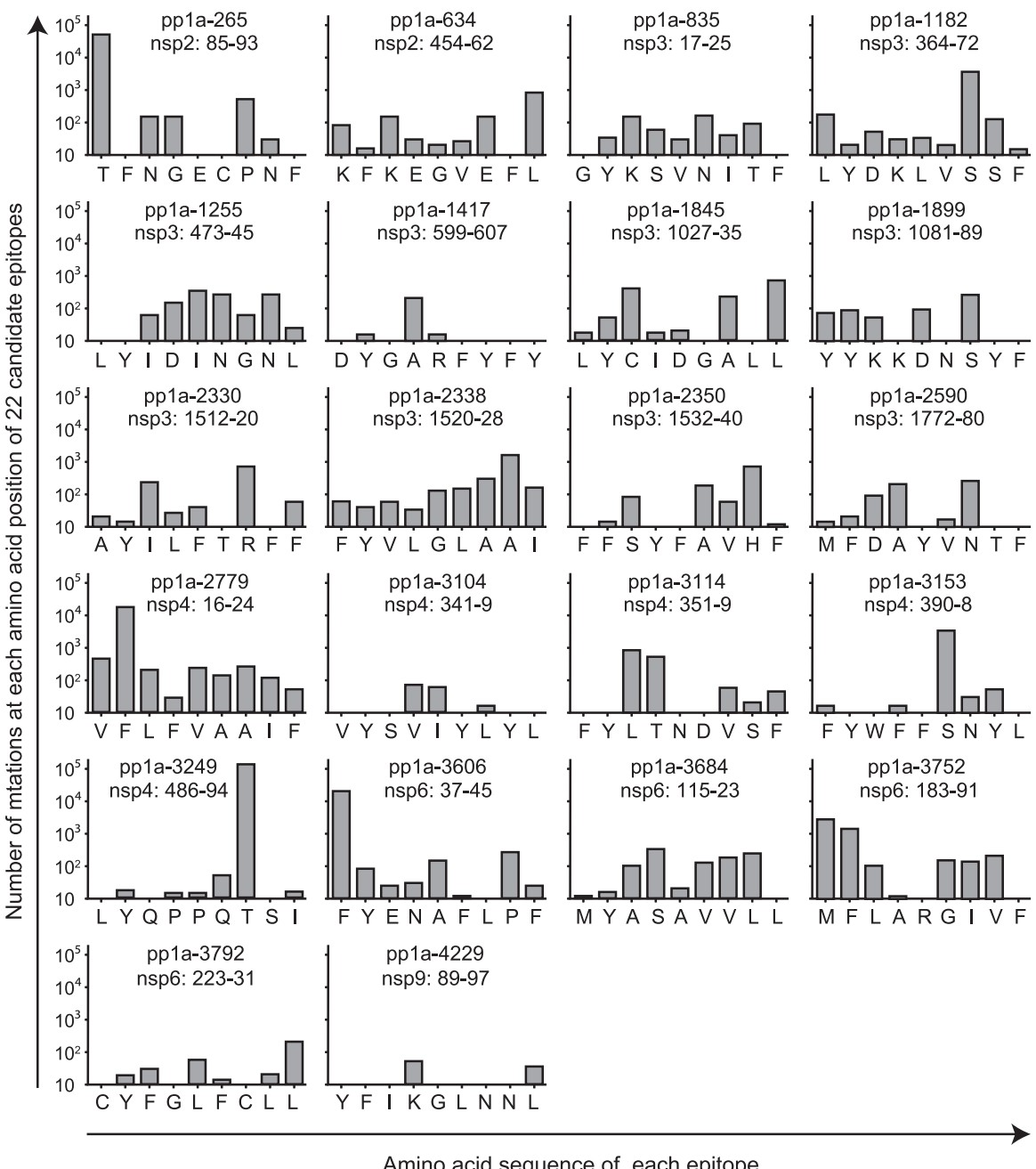

**FIG 4** Number of the total nonsynonymous mutations at each amino acid position of 22 candidate epitopes. Number of the total nonsynonymous amino acid substitutions at each amino acid position of 22 candidate epitopes was counted using the SRA data of SARS-CoV-2 variants in the NCBI Virus database.

indicated to be the most dominant epitope in the induction of activated CD8+ T cells (Fig. 5).

In the current study, we have focused on HLA-A*24:02-restricted CTL epitopes because HLA-A*24:02 is predominant in East Asian people (34, 40) such as Japanese (allele frequency: 32.7%) (41). On the other hand, HLA-A*02:01 individuals are well known to be highly frequent all over the world (34). We previously identified 18 of HLA-A*02:01-restricted CTL candidate epitopes derived from SARS-CoV-2 pp1a using HLA-A*02:01 transgenic mice (42). Then, we here examined how much these epitopes were mutated across the vast SRA data. As shown in Table 6, four epitopes involving pp1a-2785, -2884,

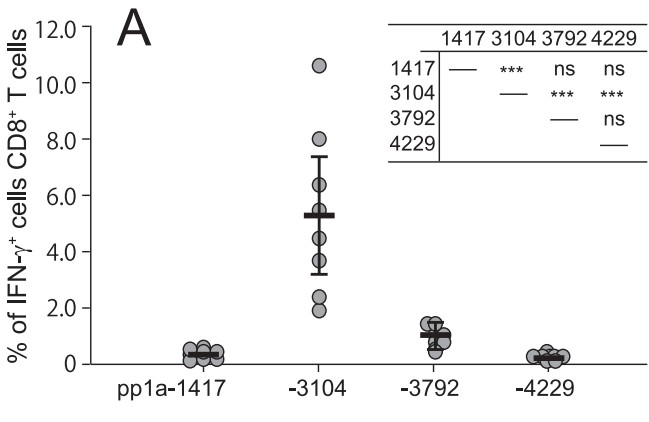

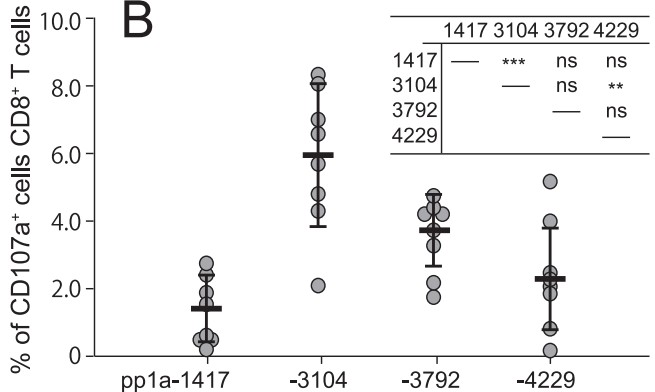

**FIG 5** Comparison of the conserved top 4 peptides in the induction of IFN-$\gamma^+$ CD8$^+$ T cells (A) and CD107a$^+$ CD8$^+$ T cells (B). Eight mice were immunized with the mixture of 4 peptides involving pp1a-1417, -3104, -3792, and -4229 in liposomes with CpG. After 1 week, spleen cells were stimulated with or without each of the 4 peptides, and the expression of intracellular IFN-$\gamma$ (A) or CD107a (B) in CD8$^+$ T cells was stained. Data indicate the relative percentages of IFN-$\gamma^+$ (A) and CD107a$^+$ (B) cells in CD8$^+$ T cells which were obtained by subtracting the % of IFN-$\gamma^+$ and CD107a$^+$ cells in CD8$^+$ T cells without a peptide from the % of IFN-$\gamma^+$ and CD107a$^+$ cells in CD8$^+$ T cells with a peptide, respectively. Each gray circle represents an individual mouse. Data are shown as the mean (horizontal bars) $\pm$ SD. Statistical analyses of the data among the 4 peptides in Fig. 5A and B were performed by one-way ANOVA followed by *post hoc* tests. Results of statistical analyses were shown as a table in the upper right corner of each figure. **, $P < 0.01$; ***, $P < 0.001$; ns, not significant.

-3403, and -3583 were found to be relatively conserved because of their low mutation frequencies per SRA run. Figure 6 indicates where the four HLA-A*02:01-restricted (Table 6), and seven HLA-A*24:02-rescricted (Table 5) epitopes with minimum mutation frequencies of less than 0.1% are located in the pp1a, indicating that these epitopes are interspersed in the five nonstructural proteins. If the nucleotide sequences encoding some of these CTL epitopes are inserted into the current mRNA vaccine or adenoviral-vectored vaccine, the new vaccine would be effective against almost all of the existing and presumably upcoming variants in HLA-A*02:01 and/or A*24:02 positive individuals who are equivalent to a significant proportion of the world's population. The new vaccine would elicit both virus-neutralizing antibodies directed against the S protein and pp1a-derived conserved epitope-specific CTLs targeting cells infected with most of the variants. Recently, Aparicio et al. (43) showed the polyepitope at region 446–480 in the receptor binding domain of S protein that elicited neutralizing antibodies cross-recognizing SARS-CoV-2 variants of concern. The peptide 446–480 contained murine CD4$^+$ and CD8$^+$ T cell epitopes as well. Hence, they suggested this polyepitope could be the basis for a peptide vaccine or other vaccine platforms such as mRNA vaccine and vectored vaccine against COVID-19. This nice idea seems to be similar to ours that induction of

**TABLE 6** Count of total nonsynonymous amino acid changes in each of the 18 HLA-A*02:01-restricted, pp1a-specific CTL candidate epitopes identified in the previous study (42)

| Name | Sequence | Protein | Position[a] | Count[b] |
|------|----------|---------|-------------|----------|
| pp1a-38 | VLSEARQHL | nsp1 | 38-46 | 920 (0.10%) |
| pp1a-52 | GLVEVEKGV | nsp1 | 52-60 | 1279 (0.14%) |
| pp1a-84 | VMVELVAEL | nsp1 | 84-92 | 8755 (0.95%) |
| pp1a-103 | TLGVLVPHV | nsp1 | 103-111 | 2103 (0.23%) |
| pp1a-445 | GLNDNLLEI | nsp2 | 265-273 | 1995 (0.22%) |
| pp1a-597 | VMAYITGGV | nsp2 | 417-425 | 2399 (0.26%) |
| pp1a-641 | FLRDGWEIV | nsp2 | 461-469 | 1867 (0.20%) |
| pp1a-1675 | YLATALLTL | nsp3 | 857-865 | 2062 (0.22%) |
| pp1a-2785 | AIFYLITPV | nsp4 | 22-30 | 741 (0.08%) |
| pp1a-2884 | FLPRVFSAV | nsp4 | 121-129 | 504 (0.05%) |
| pp1a-3083 | LLFLMSFTV | nsp4 | 320-328 | 4508 (0.49%) |
| pp1a-3403 | FLNGSCGSV | nsp5 | 140-148 | 49 (0.01%) |
| pp1a-3467 | VLAWLYAAV | nsp5 | 204-212 | 1636 (0.18%) |
| pp1a-3583 | LLLTILTSL | nsp6 | 14-22 | 194 (0.02%) |
| pp1a-3662 | RIMTWLDMV | nsp6 | 93-101 | 1463 (0.16%) |
| pp1a-3710 | TLMNVLTLV | nsp6 | 141-149 | 29228 (3.16%) |
| pp1a-3732 | SMWALIISV | nsp6 | 163-171 | 1475 (0.16%) |
| pp1a-3886 | KLWAQCVQL | nsp7 | 27-35 | 2300 (0.25%) |

[a]Amino acid position in each nonstructural protein.
[b]Count of the total amino acid substitutions present in the 9-mer amino acid sequence of each HLA-A*02:01-restricted, pp1a-specific CTL candidate epitope that were found in the SRA database of SARS-CoV-2 variants. Percentage in parenthesis indicates the mutation frequency per SRA run.

both neutralizing antibodies and T cells will generate the desired next-generation vaccine. However, they could not find any dominant human T cell epitopes in the short region 446–480 (43). Finding human T cell epitopes restricted by a variety of HLA class I & class II alleles would require searching from large proteins such as pp1a.

To identify HLA-A*24:02-restricted CTL epitopes, we utilized highly reactive HLA-A*24:02 transgenic mice (44). One reason for using MHC-I transgenic mice instead of lymphocytes of SARS-CoV-2-infected individuals is that a large number of lymphocytes are required to examine many candidates of CTL epitopes. Furthermore, when using patients' lymphocytes, we are only testing whether the peptide candidates are recognized by memory CTLs. In contrast, naive mice can be used to see if the epitope candidates are able to prime peptide-specific CTLs. This may be a better criterion to judge them as vaccine antigens. However, we have to take into account that the immunogenic variation in HLA class I transgenic mice may not be identical to that in humans because the antigen processing and presentation differ between them. In addition, we did not present data showing that viral infection in a mouse model induces T cells targeting these epitopes because liposomal peptides were used as an immunogen. Since SARS-CoV-2 does not affect murine epithelial cells in the first place, the *in vivo* CTL responses may not be relevant. Hence, there is no guarantee that the candidate epitopes identified here are real pp1a-derived epitopes that are presented by human cells during live infection with SARS-CoV-2. Recently, eight epitopes with the same amino acid sequences as pp1a-265, -1182, -1899, -2330, -3104, -3114, -3249, and -3684 (Table 5) have been submitted to the Virus Pathogen Database and Analysis Resource as HLA-A*24:02-restricted pp1a-specific CTL epitopes. Five (pp1a-265, -1182, -1899, -2330, and -3249) of them were shown to be positive in T cell assays using human lymphocytes, and therefore, they are thought to be real epitopes. Three epitopes (pp1a-3104, -3114, and -3684) of them were positive in the binding assay but negative in T cell assays, suggesting that they are not likely to be real epitopes. Then, the remaining 14 candidate epitopes in Table 5 represent new candidate epitopes that have not been previously identified.

In summary, we have identified 22 kinds of HLA-A*24:02-restricted CTL candidate epitopes derived from the pp1a of SARS-CoV-2 using computational algorithms, HLA-A*24:02 transgenic mice and the peptide-encapsulated liposomes. The conservation analysis revealed that the amino acid sequences of 7 out of the 22 epitopes were hardly affected

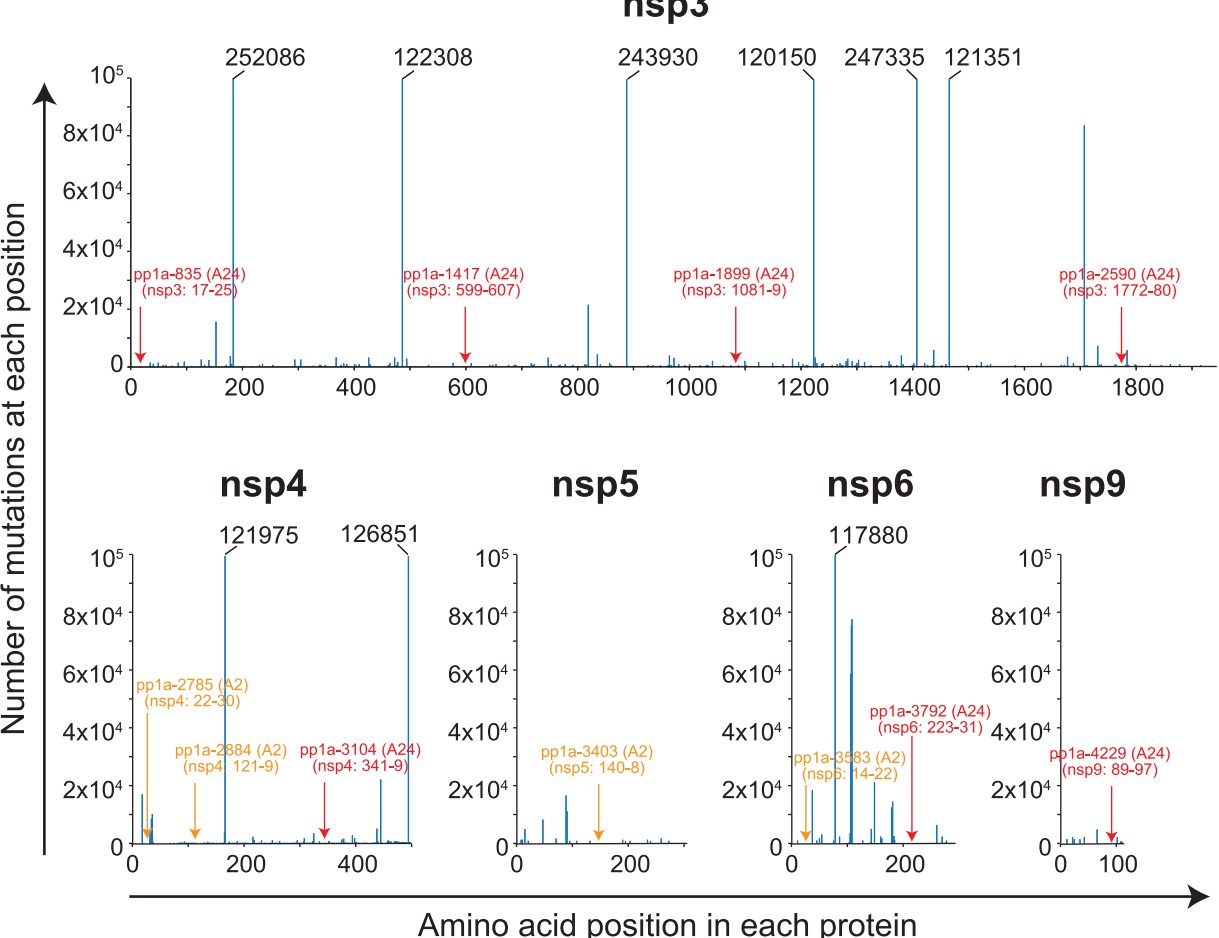

**FIG 6** Locations of conserved CTL epitopes in the pp1a. Seven HLA-A*24:02-rescricted (red letters and arrows) and four HLA-A*02:01-restricted (orange letters and arrows) epitopes were selected as conserved epitopes because they demonstrated low mutation frequencies per SRA run of less than 0.1% (Tables 5 and 6). Locations of the 11 conserved CTL epitopes were shown in this figure. The blue line indicates the number of total nonsynonymous amino acid substitutions at each amino acid position that were found in a number of SRA sequencing data of SARS-CoV-2 variants. When the number exceeds $10^5$, the actual number is shown at the top of the blue line.

by a number of mutations in the SRA database of SARS-CoV-2 variants. We also found four relatively conserved epitopes among 18 HLA-A*02:01-restricted CTL candidate epitopes that we had previously identified. The new mRNA or adenoviral-vectored vaccine containing nucleotide sequences encoding some of these epitopes might have the potential to become the universal vaccine against almost all of the existing and upcoming SARS-CoV-2 variants.

## MATERIALS AND METHODS

**Prediction of HLA-A*2402-restricted CTL epitopes.** A T-cell epitope database, SYFPEITHI (33) was used to predict HLA-A*24:02-restricted CTL epitopes derived from pp1a of SARS-CoV-2 (GenBank accession numbers: LC528232.1 & LC528233.1). Eighty of 9-mer peptides with superior scores (17 or higher) in the SYFPEITHI database were selected (Table 1) and were synthesized by Eurofins Genomics (Tokyo, Japan). These epitopes were also evaluated by three other algorithms, IEDB (34), ProPred-1 (35), and NetCTL (36) (Table 1). An HLA-A*24:02-restricted control peptide, Influenza PA$_{130-138}$ (sequence: YYLEKANKI) (37), was synthesized as well.

**Mice.** We used HLA-A*24:02 transgenic mice which were kindly provided by Dr. François A. Lemonnier (Pasteur Institute, Paris, France). The HLA-A*24:02 transgenic mouse expresses an HLA-A*24:02 monochain, designated HHD-A24, in which human $\beta$2m is covalently linked to a chimeric heavy chain composed of HLA-A*24:02 ($\alpha$1 and $\alpha$2 domains) and H-2D$^b$ ($\alpha$3, transmembrane, and cytoplasmic domains) in an H-2D$^b$, K$^b$, and mouse $\beta$2m triple knockout environment (44). Six- to 10-week-old mice were used for all experiments. Mice were housed in appropriate animal care facilities at Saitama Medical

University, and were handled according to the international guideline for experiments with animals. This study was approved by the Animal Research Committee of Saitama Medical University.

**Cell line.** The HHD-A24 gene, which was composed of human $\beta$2m cDNA linked to the chimeric heavy chain cDNA encoding $\alpha$1/$\alpha$2 domains of HLA-A*24:02, and $\alpha$3/transmembrane/cytoplasmic domains of H-2D$^b$, was synthesized by Eurofins Genomics. HHD-A24 cDNA was subcloned into the mammalian expression plasmid, pcDNA3.1 (+) (Thermo Fisher Scientific, MA) (pcDNA3.1-HHD-A24). The TAP2-dificient mouse lymphoma cell line, RMA-S (H-2$^b$) was transfected with pcDNA3.1-HHD-A24 by electroporation (Gene Pulser Xcell, Bio-Rad, Hercules, CA), and cloned by the FACSAria II cell sorter (BD Biosciences, Franklin Lakes, NJ). The resultant RMA-S-HHDA-24 cell line was cultured in RPMI 1640 medium (Nacalai Tesque Inc., Kyoto, Japan) with 10% FCS (Biowest, Nuaille, France) and 500 $\mu$g/ml G418 (Nacalai Tesque Inc.)

**Peptide binding assay.** Binding affinity of each peptide to HLA-A*24:02 was measured by the peptide binding assay using RMA-S-HHD-A24 cells, as described before (42). In brief, RMA-S-HHD-A24 cells were precultured overnight at 26°C in a $CO_2$ incubator, and pulsed with each peptide at various concentrations for 1 h at 26°C. Peptide-pulsed cells were incubated for 3 h at 37°C, and were stained with anti-HLA-A24 monoclonal antibody (MAb), A11.1 M (45), followed by FITC-labeled goat anti-mouse IgG antibody (Sigma-Aldrich, St. Louis, MO). Mean fluorescence intensity (MFI) of HLA-A*24:02 expression on the surface of RMA-S-HHD-A24 cells was measured by flow cytometry (FACSCanto II, BD Biosciences), and standardized as the percent cell surface expression by the following formula: % relative binding = [{(MFI of cells pulsed with each peptide) − (MFI of cells incubated at 37°C without a peptide)}/{(MFI of cells incubated at 26°C without a peptide) − (MFI of cells incubated at 37°C without a peptide)}] × 100. The concentration of each peptide that yields the 50% relative binding was calculated as the half-maximal binding level ($BL_{50}$).

**Peptide-encapsulated liposomes.** Peptide-encapsulated liposomes were prepared using Lipocapsulater FD-U PL (Hygieia BioScience, Osaka, Japan), as previously described (42). Briefly, each of synthetic peptides was dissolved in DMSO at a final concentration of 10 mM. For the first screening of HLA-A*24:02-restricted epitopes, 20 $\mu$l each of 6 peptide solutions was mixed together, and the total volume was increased to 2 ml by adding $H_2O$. For the identification of dominant epitopes, 20 $\mu$l each of 10 mM peptides selected was mixed together, and diluted to 2 ml with $H_2O$. The peptide solution was added into a vial of Lipocapsulater containing 10 mg of dried liposomes, and incubated for 15 min at room temperature. The resultant solution contains peptide-encapsulated liposomes.

**Immunization.** Mice were immunized s.c. four times at a 1-week interval with peptide-encapsulated liposomes (100 $\mu$l/mouse for priming and 50 $\mu$l/mouse for boosting) together with CpG-ODN (5002: 5′-TCCATGACGTTCTTGATGTT-3′, Hokkaido System Science, Sapporo, Japan) (5 $\mu$g/mouse) in the footpad.

**Intracellular cytokine staining (ICS).** ICS was performed as described previously (42). Spleen cells of immunized mice were incubated with 50 $\mu$M each peptide for 5 h at 37°C in the presence of brefeldin A (GolgiPlug, BD Biosciences), and were stained with FITC-conjugated anti-mouse CD8 MAb (BioLegend, San Diego, CA). Cells were then fixed, permeabilized, and stained with phycoerythrin (PE)-conjugated rat anti-mouse IFN-$\gamma$ MAb (BD Biosciences). After washing the cells, flow cytometric analyses were performed using flow cytometry (FACSCanto II, BD Biosciences).

**Conservation analysis of CTL epitopes.** To examine the conservation of the CTL candidate epitopes, we utilized the SRA data of SARS-CoV-2 variants in the NCBI Virus database. We counted the total number of nonsynonymous amino acid changes present in the 9-mer amino acid sequence of each epitope that were found in the SRA mutation database, and calculated percentage of the mutation frequency per SRA run of each epitope.

**Detection of CD107a molecules on CD8$^+$ T cells.** For the detection of CD107a, spleen cells of immunized mice were incubated with 50 $\mu$M each peptide for 6 h at 37°C in the presence of monensin (GolgiStop, BD Biosciences) and 0.8 $\mu$g of FITC-conjugated anti-mouse CD107a MAb (BioLegend). Cells were then stained with PE-Cy5-conjugated anti-mouse CD8 MAb (BioLegend), and were analyzed by flow cytometry (FACSCanto II, BD Biosciences).

**Statistical analyses.** One-way ANOVA followed by *post hoc* tests was performed for statistical analyses among multiple groups using GraphPad Prism 5 software (GraphPad software, San Diego, CA). A value of $P < 0.05$ was considered statistically significant.

## ACKNOWLEDGMENTS

The authors are grateful to François A. Lemonnier (Pasteur Institute, Paris, France) for providing HLA-A*24:02 transgenic mice. The authors also thank T. Nakatsura (National Cancer center, Japan) for his help of preparing HLA-A*24:02 transgenic mice. This work was supported by a Grant-in-Aid for Scientific Research (C) (JSPS KAKENHI Grant Number: JP18K06631) to M.M., and a Grant-in-Aid for Early-Career Scientists (JSPS KAKENHI Grant Number: JP18K15430) to A.T. from Japan Society for the Promotion of Science.

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
