## [Reviewer comments · Microbiology Spectrum]

Microbiology Spectrum

Identification of HLA-A*24:02-restricted CTL candidate epitopes derived from the non-structural polyprotein 1a of SARS-CoV-2 and analysis of their conservation using the mutation database of SARS-CoV-2 variants

Akira Takagi and Masanori Matsui

Corresponding Author(s): Masanori Matsui, Saitama Medical University

Review Timeline:

Submission Date:	September 23, 2021
Editorial Decision:	November 8, 2021
Revision Received:	November 17, 2021
Accepted:	November 24, 2021

Editor: Yongjun Sui

Reviewer(s): The reviewers have opted to remain anonymous.

Transaction Report:

DOI: <https://doi.org/10.1128/spectrum.01659-21>

November 8, 2021

Dr. Masanori Matsui
Saitama Medical University
Microbiology
38 Morohongo
Moroyama-cho
Iruma-gun, Saitama 350-0495
Japan

Re: Spectrum01659-21 (Identification of HLA-A*24:02-restricted CTL candidate epitopes derived from the non-structural polyprotein 1a of SARS-CoV-2 and analysis of their conservation using the mutation database of SARS-CoV-2 variants)

Dear Dr. Masanori Matsui:

Thank you for submitting your manuscript to Microbiology Spectrum. When submitting the revised version of your paper, please provide (1) point-by-point responses to the issues raised by the reviewers as file type "Response to Reviewers," not in your cover letter, and (2) a PDF file that indicates the changes from the original submission (by highlighting or underlining the changes) as file type "Marked Up Manuscript - For Review Only". Please use this link to submit your revised manuscript - we strongly recommend that you submit your paper within the next 60 days or reach out to me. Detailed information on submitting your revised paper are below.

Link Not Available

Sincerely,

Yongjun Sui

Journals Department
Reviewer comments:

Reviewer #1 (Comments for the Author):

Concerns and Suggestions:

1. The work presented by Takagi, A. is mostly based on a computational algorithm-based identification of conserved epitope in evolving SARS-CoV-2 variants, and their studies indicate that a few of those epitopes have the ability to induce significantly higher CTL immune response. However, it would be important to show corresponding S protein neutralizing antibody production in the test mice.
2. Please refer and discuss relevant recent work directed to identify the epitopes for subunit peptide vaccine using a multiepitopic and cross-reactive humoral neutralizing and cellular CD4 and CD8 response strategy against COVID19 (Aparicio, et al., Emerg Microbes Infect. 2021 Dec; 10(1):1931-1946.)
3. Please simplify figure 3 by adding unified statistical data (as in Table 4) in graph format and showing representative flow cytometry histograms only.

Reviewer #2 (Comments for the Author):

One major concern this reviewer has is about relevance of model to humans. The analysis of SARS-CoV-2 peptides that authors developed and tested in mice may not be relevant to humans. SARS-CoV-2 does not affect murine epithelial cells, so the in vivo CTL responses may not be relevant. Authors must explain or clear this point. Authors can also choose to test these peptides using relevant model system

Staff Comments:

Preparing Revision Guidelines

Please return the manuscript within 60 days; if you cannot complete the modification within this time period, please contact me. If you do not wish to modify the manuscript and prefer to submit it to another journal, please notify me of your decision immediately so that the manuscript may be formally withdrawn from consideration by Microbiology Spectrum.

Thank you for inviting me to review the manuscript entitled, "Identification of HLA-A*24:02-restricted CTL candidate epitopes derived from the non-structural polyprotein 1a of SARS-CoV-2 and analysis of their conservation using the mutation database of SARS-CoV-2 variants" by Takagi, A., and Matsui, M.

This work focused on understating and solving an important aspect of enhanced transmissibility and/or immunoevasion by the novel variants to escape from vaccine-elicited immunity. In addition to tradition antibody response, the work is dedicated to explore potential precision anti SARS-CoV-2 spike protein immune responses of CD8⁺ cytotoxic T lymphocytes (CTLs). The authors identified twenty-two HLA-A*24:02-restricted CTL candidate epitopes derived from the non-structural polyprotein 1a (pp1a) of SARS-CoV-2 using computational algorithms, HLA-A*24:02 transgenic mice and the peptide-encapsulated liposomes. This study revealed that the amino acid sequences of 7 out of the 22 epitopes were hardly affected by a number of mutations in the Sequence Read Archive database of SARS-CoV-2 variants and predicted that these conserved epitopes might be useful for designing the next-generation COVID-19 vaccine that is universally effective against any SARS-CoV-2 variants by the induction of both anti-Spike neutralizing antibodies and CTLs specific for conserved epitopes. This work highlights aspects of designing the next-generation COVID-19 vaccine in order to compensate for the viral evolution that might be universally effective against the new SARS-CoV-2 variants.

Concerns and Suggestions:

1. The work presented by Takagi, A. is mostly based on computational algorithm based identification of conserved epitope in evolving SARS-CoV-2 variants, and their studies indicate that a few of those epitopes have the ability to induce significantly higher CTL immune response. However, it would be important to show corresponding S protein neutralizing antibody production in the test mice.
2. Please refer and discuss relevant recent work directed to identify the epitopes for subunit peptide vaccine using a multiepitopic and cross-reactive humoral neutralizing and cellular CD4 and CD8 response strategy against COVID19 (Aparicio, et al., *Emerg Microbes Infect.* 2021 Dec; 10(1):1931-1946.)
3. Please simplify figure 3 by adding a unified statistical data (as on Table 4) in graph format and show representative flow cytometry histograms only.

Response to Reviewers

Manuscript number: Spectrum01659-21

1) Reviewer #1

Thank you very much for the very helpful comments and suggestions.

1. We feel that this comment of Reviewer #1 results from a misinterpretation of the data. In this paper, mice were immunized with liposomal 9-mer short peptides derived from the non-structural polyprotein 1a (pp1a) of SARS-CoV-2 for the induction of pp1a-specific CTLs. Because mice were not immunized with S protein or S protein-derived peptides, anti-S protein neutralizing antibodies were not produced in the test mice. We just suggested in the discussion (Pages 15-16, lines 288-294) that if the nucleotide sequences encoding pp1a-derived CTL epitopes are inserted into the current mRNA vaccine or vectored vaccine, the new vaccine would be very effective for any variants because of the induction of both anti-S protein neutralizing antibodies and pp1a-specific CTLs.

2. We are very grateful to Reviewer #1 for suggesting the addition of a good reference. We have referred (reference #43, Page 34, lines 627-632) and discussed (Page 16, lines 294-303) about this paper (Emerg Microbes Infect 10:1931-1946).

Page 16, lines 294-303: “Recently, Aparicio et al. (43) showed the polyepitope at region 446-480 in the receptor binding domain of S protein that elicited neutralizing antibodies cross-recognizing SARS-CoV-2 variants of concern. The peptide 446-480 contained murine CD4⁺ and CD8⁺ T cell epitopes as well. Hence, they suggested this polyepitope could be the basis for a peptide vaccine or other vaccine platforms such as mRNA vaccine and vectored vaccine against COVID-19. This nice idea seems to be similar to ours that induction of both neutralizing antibodies and T cells will generate the desired next-generation vaccine. However, they could not find any dominant human T cell epitopes in the short region 446-480 (43). Finding human T cell epitopes restricted by a variety of HLA class I & class II alleles would require searching from large proteins such as pp1a.”

3. We agree with the referee in this regard. We have simplified Fig. 3 by adding unified statistical data (Fig. 3A) in graph format and have shown representative flow cytometry histograms as Fig. 3B.

Accordingly,

Page 10, line 191: “Fig. 3” has been changed to “Fig. 3A”.

Page 11, line 199: “(Fig. 3A & B)” has been added.

Page 11, line 200: “Fig. 3” has been changed to “Fig. 3A”.

Page 35, lines 654-660: We have added the following sentences in fig. legends.

Page 35, lines 654-660: “(A) Values of ICS show the relative percentages of IFN- γ ⁺ cells in CD8⁺ T cells which were calculated by subtracting the % of IFN- γ ⁺ cells in CD8⁺ T

cells without a peptide from the % of IFN- γ ⁺ cells in CD8⁺ T cells with a relevant peptide. Thirty-six peptides tested were divided into 4 groups with ICS values of 1% or higher, 0.5-1%, 0.1-0.5%, and ND (not detected). (B) Representative flow cytometry histograms are shown. Numbers shown indicate the percentages of intracellular IFN- γ ⁺ cells within CD8⁺ T cells stimulated with (+) or without (-) a relevant peptide.”

2) Reviewer #2

Thank you very much for the very helpful comments.

We absolutely agree with Reviewer #2 in this point. As Reviewer #2 mentioned, the analysis of SARS-CoV-2 peptides that we developed and tested in mice may not be relevant to humans. However, we already described several sentences about this issue in the discussion as follows.

Pages 16-17, lines 311-315: “However, we have to take into account that the immunogenic variation in HLA class I transgenic mice may not be identical to that in humans because the antigen processing and presentation differ between them. In addition, we did not present data showing that viral infection in a mouse model induces T cells targeting these epitopes because liposomal peptides were used as an immunogen.

Page 17, lines 316-318: “Hence, there is no guarantee that the candidate epitopes identified here are real pp1a-derived epitopes that are presented by human cells during live infection with SARS-CoV-2.”

Therefore, we do not think we need to write more about this issue. However, according to the comment of Reviewer #2, we have added the following new sentence.

Page 17, lines 315-316: “Since SARS-CoV-2 does not affect murine epithelial cells in the first place, the *in vivo* CTL responses may not be relevant.”

As Reviewer #2 suggested in the last sentence of this comment, we will check these peptides with human cells in the near future.

3) A minor point:

We have added the following sentence in the acknowledgements.

Page 24, lines 433-434: “The authors also thank Dr. T. Nakatsura (National Cancer center, Japan) for his help of preparing HLA-A*24:02 transgenic mice.”

November 24, 2021

Dr. Masanori Matsui
Saitama Medical University
Microbiology
38 Morohongo
Moroyama-cho
Iruma-gun, Saitama 350-0495
Japan

Re: Spectrum01659-21R1 (Identification of HLA-A*24:02-restricted CTL candidate epitopes derived from the non-structural polyprotein 1a of SARS-CoV-2 and analysis of their conservation using the mutation database of SARS-CoV-2 variants)

Dear Dr. Masanori Matsui:

Your manuscript has been accepted, and I am forwarding it to the ASM Journals Department for publication. You will be notified when your proofs are ready to be viewed.

Sincerely,

Yongjun Sui
Editor, Microbiology Spectrum
